# INTEGRATIVE TENSOR-BASED ANOMALY DETECTION SYSTEM FOR SATELLITES

## ABSTRACT

Detecting anomalies is of growing importance for various industrial applications and mission-critical infrastructures, including satellite systems. Although there have been several studies in detecting anomalies based on rule-based or machine learning-based approaches for satellite systems, a tensor-based decomposition method has not been extensively explored for anomaly detection. In this work, we introduce an Integrative Tensor-based Anomaly Detection (ITAD) framework to detect anomalies in a satellite system. Because of the high risk and cost, detecting anomalies in a satellite system is crucial. We construct 3rd-order tensors with telemetry data collected from Korea Multi-Purpose Satellite-2 (KOMPSAT-2) and calculate the anomaly score using one of the component matrices obtained by applying CANDECOMP/PARAFAC decomposition to detect anomalies. Our result shows that our tensor-based approach can be effective in achieving higher accuracy and reducing false positives in detecting anomalies as compared to other existing approaches.

## 1 INTRODUCTION

Due to the high maintenance cost as well as extreme risk in space, detecting anomalies in a satellite system is critical. However, anomaly detection in a satellite system is challenging for several reasons. First, anomalies occur due to complex system interactions from various factors inside and outside a satellite system. For example, a sensor in one subsystem in a satellite system is often connected to several other types of sensors or resources in other subsystem modules. Each sensor measurement is encapsulated as telemetry and downlinked to the ground station. In order to identify anomalies, it is crucial to compare and understand not just one single telemetry but several telemetries as a whole. However, most of the previous studies (Fuertes et al., 2016; Hundman et al., 2018; OMeara et al., 2016) on detecting anomalies in satellite systems have primarily focused on analyzing individual telemetry. This can lead to a high false positives rate, because some instantaneous glitches may not be actual anomalies, but just trivial outliers (Yairi et al., 2017). Additionally, false positives can be costly, requiring much manual effort from operators to investigate and determine whether they are anomalies (Hundman et al., 2018). To reduce the false positives, analyzing a set of multiple telemetries as a whole can be more effective to determine true anomalies in a complex system. To the best of our knowledge, this integrated approach for a satellite system has not been studied extensively in the past.

In order to address these challenges, we propose an Integrative Tensor-based Anomaly Detection (ITAD) framework for a satellite system, where a tensor can effectively capture a set of high dimensional data. Specifically, we construct a 3rd-order tensor for entire telemetries in one subsystem and decompose it into component matrices, which captures the characteristics of multiple telemetries as a whole to detect anomalies. We then conduct a cluster analysis on one component matrix in a decomposed tensor and calculate the anomaly score based on the distance between each telemetry sample and its cluster centroid. Finally, we used the dynamic thresholding method (Hundman et al., 2018) to detect anomalies; the dynamic thresholding method changes the detection threshold value over time instead of using a fixed value for the entire dataset. We performed experiments on our approach with a subset of real telemetries from the KOMPSAT-2 satellite, and verify that our approach can detect actual anomalies effectively and reduce false positives significantly, compared to other approaches.

## 2 RELATED WORK

The fundamental concept of a tensor and the algorithm for tensor decomposition are described in Appendix A. In this section, we mainly provide an overview of the research that is directly relevant to our work.

**Tensor-based Anomaly Detection:** Tensor decomposition for anomaly detection has been developed by Nomikos and MacGregor (1994). They proposed multiway principal components analysis (MPCA) using tensor decomposition to monitor the progress of the batch process in the multivariate trajectory data. They successfully extracted information from a database of batches and captured the multivariate data into a few matrices, which can represent the original data sufficiently, while reducing the space. The modern application of tensor-based anomaly detection has been deployed in many different areas such as neuroscience, environmental monitoring, video surveillance, network security, and remote sensing.

The tensor-based anomaly detection method can detect anomalies in an unsupervised manner, where the score plot-based model is the most widely used tensor decomposition method in anomaly detection without labels. Component matrices from tensor decomposition are utilized to calculate the score plots to detect the anomalies. According to the characteristics of the tensor dataset, the score plot can be 1-dimensional (Kosanovich et al., 1994; Gauvin et al., 2014; Papalexakis et al., 2014), 2-dimensional (Cong et al., 2013; Lee et al., 2014), or 3-dimensional (Mao et al., 2014; Fanaee-T and Gama, 2014). In our work, we adopt a score-based method to detect anomalies using one of the component matrices, which has the comprehensive characteristics of all telemetries simultaneously along the timeline.

**Anomaly Detection for Satellite Operation:** Generally, the Out-Of-Limit (OOL) is one of the most widely used methods to detect an anomaly, where OOL can define a nominal range with the lower and upper thresholds. So far, detecting anomalies for satellite systems have primarily used the OOL along with additional methods such as dimensionality reduction algorithm (Schölkopf et al., 1998; Fujimaki et al., 2005; Inui et al., 2009), nearest neighbors (Breunig et al., 2000; Bay and Schwabacher, 2003; Iverson, 2008; Kriegel et al., 2009), and clustering algorithm (Iverson, 2004; Gao et al., 2012; Li et al., 2017).

Recently, machine learning-based approaches have been studied in order to automate the anomaly detection process. Centre National d'Etudes Spatials (CNES) proposed a telemetry monitoring method, NOSTRADAMUS (Fuertes et al., 2016), which transforms the original telemetry dataset into a vector matrix of features and reduces the dimensions using Principal Component Analysis (PCA). After that, One-class Support Vector Machine (OCSVM) is applied with a decision frontier to detect anomalies. On the other hand, Yairi et al. (2017) proposed a data-driven health monitoring and anomaly detection method for the Japan Aerospace Exploration Agency (JAXA), based on probabilistic dimensionality reduction and clustering. They proposed the Mixtures of Probabilistic Principal Component analyzers (MPPCA) (Tipping and Bishop, 1999) and Categorical Distribution (CD). Parameters of analyzers are determined using the EM algorithm.

Furthermore, the Automated Telemetry Health Monitoring System (ATHMoS) is developed by the German Space Operation Center (GSOC) (OMeara et al., 2016; 2018). They used an autoencoder to obtain compressed data representing the original dataset efficiently. They show that the feature vector extracted for the autoencoder is utilized for anomaly detection and decrease the false positives. Also, the National Aeronautics and Space Administration (NASA)'s Jet Propulsion Laboratory utilizes Long Short-Term Memory (LSTM) networks with label information provided by an expert to detect anomaly for their MSL and SMAP spacecraft (Hundman et al., 2018). They smooth the sharp spikes in error value using an exponentially-weighted average (EWMA). They also proposed an unsupervised model and false-positive mitigation method. However, all the prior research focused on detecting anomaly from individual telemetry, rather than from a set of telemetries.

## 3 DATASET

Korea Aerospace Research Institute (KARI) (KARI, 2019) is the aeronautics and space agency in South Korea founded for the development and research of aerospace scientific technologies. KARI has developed and is currently operating a series of multipurpose satellites for high-resolution optical observation, and radar and IR observation, as well as geostationary orbit satellites for indepen-

dent weather and marine observation, and communication relay. Korea Multi-Purpose Satellite 2 (KOMPSAT-2) (KOMPSAT-2, 2019) is one of the national monitoring satellites with high-resolution optical observation. It was launched in 2006 and had been operating for 13 years. KOMPSAT-2 satellite generates more than 3,000 different types of telemetries from various subsystems (Lee et al., 2005; eoPortal, 2019).

In this work, we collected 88 types of different telemetries with more than 43,716 telemetry samples from the KOMPSAT-2 satellite for 10 months. These telemetries are categorized into 7 different subsystems according to their characteristics. In Table 1, we present the number of different telemetry types used from each subsystem collected from May 2013 to February 2014 for 10 months. Also, the collected data size for each month is shown in GB.

Table 1: Telemetry dataset description from KOMPSAT-2 satellite collected from *May 2013* till *Feb. 2014*.

| Subsystem | Sub1 | Sub2 | Sub3 | Sub4 | Sub5 | Sub6 | Sub7 | Total |
|---|---|---|---|---|---|---|---|---|
| Num. of different telemetry types | 7 | 9 | 14 | 26 | 15 | 10 | 7 | 88 |
| Data size (GB) | 1.15 | 1.24 | 1.40 | 1.83 | 1.26 | 1.44 | 1.20 | 9.52 |

## 4 INTEGRATIVE TENSOR-BASED ANOMALY DETECTION (ITAD)

In this section, we explain the overall process of our new approach, an Integrative Tensor-based Anomaly Detection (ITAD) framework for the satellite system and describe its details in Fig. 1.

**Data Interpolation and Normalization (Preprocessing):** Since many telemetries are measured and sampled at a different time interval (from every second to a few minutes), there are many missing or unmeasured sensor values for each timestamp. To address this challenge, we apply linear interpolation, which populates a missing value with a linearly increasing value between two consecutive data points. After linear interpolation, we normalize each telemetry value (feature) individually. Each value such as temperature or power is measured on a different scale; min-max normalization is used to normalize all values into the range between 0 and 1. The minimum value is mapped to 0, and the maximum value is mapped to 1.

After linear interpolation, the timestamp is recorded every 1 second in the raw dataset. However, most of the values do not change in a short period, and many telemetry values have the same value for a few minutes or much longer. Therefore, we assumed that it might be practical to compress several timestamps (rows) into a single row. Also, as we add many interpolated values from the linear interpolation step, the size of the dataset increases by more than three times (9.52GB to 29.62GB) after interpolation. Therefore, the compression provides the benefit in computational efficiency while maintaining the critical information of the data.

**Feature Extraction:** Using the above compression method, we use different statistical methods to extract 8 features for each telemetry time series $T_i$: mean ($\bar{x}$), standard deviation ($s$), skewness ($skew$), kurtosis ($kurt$), minimum ($min$), maximum ($max$), energy ($E$), and average crossing ($x_{crossing}$). Energy and average crossing are calculated from $E = \frac{1}{N} \sum_{i=1}^{N} x_i^2$ and $\bar{x}_{crossing} = \frac{1}{N} \sum_{i=1}^{N} 1_{x_i > \bar{x}}$, respectively. Each feature is calculated for every 10 minutes of the

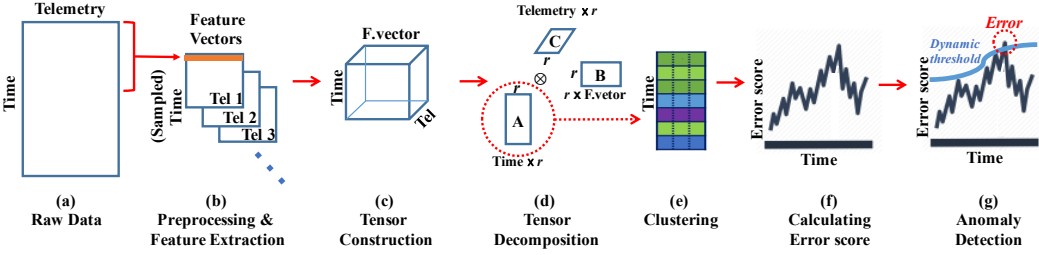

Figure 1: The end-to-end data processing pipeline of our Integrative Tensor-based Anomaly Detection (ITAD) approach.

original data and we obtain the final feature vector $V_{\{n,T_i\}}$ generated by concatenating the different features as shown in equation 1 as follows:

$$V_{\{n,T_i\}} = \{\bar{x}, s, skew, kurt, min, max, E, \bar{x}_{crossing}\}, \tag{1}$$

where $n$ is the number of feature vector samples. As a result, we can reduce the size of the dataset significantly from the interpolated data (0.5 Gb $<<$ 29.62 GB) and reconstitute the telemetry dataset into the matrix form consisting of feature vectors (at each column) by time samples (at each row) as shown in Fig. 1.(b).

**Tensor Construction and Decomposition:** Tensor decomposition can effectively handle high dimensional time series. Therefore, we construct a 3rd-order telemetry tensor consisting of $time \times feature\ vector \times telemetry$ and decompose a tensor using the CANDECOMP/PARAFAC (CP) decomposition in Fig. 1.(c), which is one of the most widely used tensor decomposition methods, as shown in equation 2. After CP decomposition, we obtain three component matrices, $A$, $B$, and $C$ as described in Fig. 1.(d). The component matrix $A$ consists of time-to-factor (time $\times$ $r$) describing the comprehensive characteristics of samples from all telemetries at the same point in time using $r$ factors. The component matrix $B$ shows the feature vector-to-factor matrix ($r \times$ feature vector) indicating how much each factor influences each feature vector. The final matrix $C$ captures the telemetry-to-factor (telemetry $\times$ $r$) matrix to characterize how much each factor affects each telemetry.

$$\boldsymbol{X} \approx \sum_{r=1}^{R} \lambda_r a_r \circ b_r \circ c_r = [\![\lambda; \boldsymbol{A}, \boldsymbol{B}, \boldsymbol{C}]\!], \tag{2}$$

where $R$ is the rank, $\lambda$ is the weight, $a_r \in \mathbb{R}^{I_1}, b_r \in \mathbb{R}^{I_2}$, and $c_r \in \mathbb{R}^{I_3}$, for $r$=1, ..., R (Hackbusch, 2012).

In order to find the optimal solutions of CP decomposition, we utilize the alternating least squares (ALS) updating method, which iteratively optimizes one component while leaving the others fixed. Given the 3rd-order tensor, it first fixes the component matrices $B$ and $C$ to obtain the solution for the component matrix $A$. Then, ALS fixes $A$ and $C$ to find $B$, and lastly fixes $A$ and $B$ to solve for $C$ as follows: $\boldsymbol{A} \leftarrow arg\ min_A \|\boldsymbol{X}_{(1)} - \boldsymbol{A}(\boldsymbol{C} \odot \boldsymbol{B})^\top\|$, $\boldsymbol{B} \leftarrow arg\ min_B \|\boldsymbol{X}_{(2)} - \boldsymbol{B}(\boldsymbol{C} \odot \boldsymbol{A})^\top\|$, $\boldsymbol{C} \leftarrow arg\ min_C \|\boldsymbol{X}_{(3)} - \boldsymbol{C}(\boldsymbol{B} \odot \boldsymbol{A})^\top\|$, where $\boldsymbol{X}_{(1)}$ denotes the mode-1 unfolding of tensor $\boldsymbol{X}$ into a matrix, and $\boldsymbol{X}_{(2)}$ and $\boldsymbol{X}_{(3)}$ indicates the mode-2 and mode-3 unfolding, respectively. Moreover, $\odot$ denotes the *Khatri-Rao product* (Smilde et al., 2004), which is the "matching columnwise" Kronecker product. It repeats this procedure until it reaches the specific convergence criteria or maximum iteration.

**Selecting an Optimal Rank *r*:** Since we aim to obtain the component matrices from the decomposition, it is critical to choose an optimal size of the factor (rank) $r$ that can represent the original telemetry tensor. However, there is no general straightforward algorithm to select the optimal $r$. Instead, we measure the reconstruction error as the difference between the original tensor $\boldsymbol{X}$ and the approximated tensor $\hat{\boldsymbol{X}}$. From the given 3rd-order tensor $\boldsymbol{X} \in \mathbb{R}^{I_1 \times I_2 \times I_3}$, we use the Frobenius norm $\|\boldsymbol{X} - \hat{\boldsymbol{X}}\|_{\mathbf{F}}$ to calculate the reconstruction error, where $\hat{\mathbf{X}}$ is computed as the outer product ($\circ$) of component matrices $A$, $B$, and $C$, as shown in equation 2. We can compute the reconstruction error as follows: $Reconstruction\ error = \sum_{ijk}(x_{ijk} - \sum_{r=1}^{R} a_{ir}b_{jr}c_{kr})^2$, for $i = 1, ..., I, j = 1, ..., J, k = 1, ..., K, r = 1, ..., R$, where $x_{ijk} \in \mathbb{R}^{I \times J \times K}$, $a_{ir} \in \mathbb{R}^{I \times R}$, $b_{jr} \in \mathbb{R}^{J \times R}$, and $c_{kr} \in \mathbb{R}^{K \times R}$. The smaller the reconstruction error, the closer the approximate tensor is to the original tensor. We find the reconstruction error by increasing the rank $r$ from 2 and choosing the smallest $r$ by minimizing $\|\boldsymbol{X} - \hat{\boldsymbol{X}}\|_{\mathbf{F}}$, until when the approximated tensor can reconstruct more than 90% of the original telemetry tensor. We present an example of selecting an optimal $r$ from the reconstruction error in Appendix B.

**Clustering Analysis:** The original telemetry data is highly unbalanced, where most elements are normal, with only a few anomalies. This is one of the challenging issues in an anomaly detection problem. Additionally, normal telemetry data might exhibit certain repeating patterns, as many satellite commanding sequences are not drastically different from one another during nominal operation.

Therefore, we apply a clustering method for the component matrix $A$ ($Time \times r$), in order to group major patterns of telemetry samples, as shown in Fig. 1.(e), such that they represent normal data behavior. The primary reason we chose the matrix $A$ among the three-component matrices

is that we ultimately aim to identify the time, at which the anomaly occurs, and the component matrix $A$ represents the key information and comprehensive characteristics of all telemetries at each time instance across different subsystems. Note that the clustering is applied row-wise since the component matrix $A$ has the time information at its row.

However, clustering is challenging because not only a telemetry sample is an 8-dimensional vector, but also the original tensor dataset consists of different types of telemetries. Therefore, we extensively experimented with several clustering algorithms such as Gaussian Mixture Model (GMM), Isolation Forest (IF), $k$-means, and One-class Support Vector Machine (OCSVM) to compare and determine the best approach. Clustering methods other than the $k$-means algorithm showed too many false positives. Hence, we only use the $k$-means algorithm in our work.

As it is required to set the number of clusters when applying $k$-means clustering, we use silhouette analysis to determine an optimal $k$. The silhouette method (Rousseeuw, 1987) has coefficients ranging from -1 to 1, where the positive value indicates that the samples are correctly assigned and away from the neighboring clusters, and the negative value represents samples that might be assigned to the wrong cluster. The zero value indicates that the samples are on the decision boundary of two neighboring clusters. We varied $k$ from 2 to 10 and chose the value when the silhouette coefficient is the largest positive value, as shown in Appendix B.

Note: Tensor decomposition can be viewed as a clustering mechanism. Generally, the component matrix $A$ ($time \times factor$), which accounts for comprehensive characteristics of all telemetries in the same subsystem, can serve as an indicator for different column-wise clusters. In our research, however, we need a row-wise clustering for calculating an anomaly score by time, since our goal is to identify the time instance when an anomaly occurs. That is the reason we use another $k$-means clustering in addition to the tensor decomposition to capture the distance between normal and abnormal data.

**Calculating Anomaly Score:** If a time sample is normal, it might belong to one of the clusters we constructed from the previous step. If a time sample is anomalous, then it would exhibit a far different pattern from normal patterns, and it would not belong to any clusters. To quantify this, we calculate the Euclidean distance $d(\mathbf{s}, \mathbf{c}) = \sqrt{\sum_{i_1=1}^{n}(c_i - s_i)^2}$ between each time sample $\mathbf{s} = (s_1, s_2, ..., s_n)$ and the centroid $\mathbf{c} = (c_1, c_2, ..., c_n)$ of its nearest cluster. A short Euclidean distance means that a value is similar to a normal value and pattern, and a long distance indicates that the value is far different from major clusters and normal patterns. Therefore, we can define this Euclidean distance as an anomaly score, as shown in Fig. 1.(f), where anomalies will typically have high anomaly scores.

**Data-Driven Dynamic Thresholding:** In order to derive anomalies from the anomaly score, it is required to set a certain threshold. Although a fixed threshold is the simplest way, it cannot detect the contextual anomalies, which can be below the threshold point. In the same vein, values that are just above the fixed threshold can be normal, but they can be detected as anomalies with a fixed threshold method. Additionally, a fixed threshold approach cannot adapt to various changing conditions and can produce too many false positives. The example of the problems with a fixed threshold and high false positives is illustrated in Appendix C. To address the problem with a fixed threshold, we develop the data-driven dynamic thresholding method, where a threshold value can be dynamically adjusted and changed adaptively in differing contexts. We first choose the time window $w$, which is defined as the number of previous anomaly score points to compute the current anomaly score. Then, we calculate the mean $\mu$ and standard deviation $\sigma$ of the data values in the time window $w$. Finally, based on the confidence interval distribution, we determine an anomaly, when the anomaly score is over $m \times \sigma$ apart from the $\mu$, denoted by $\sigma = \sum(X_i - \mu)^2$ and $\mu = \sum X_i/n$, where $i = (n - w), ..., n$ (the number of data points in $w$) and $m$ (the coefficient parameter) $\geq 1$. This measures how far ($m \times \sigma$) is apart from the mean $\mu$ of values within the window $w$.

## 5 EXPERIMENT

**Tensor Size:** In this experiment, we use 88 types of telemetries with more than 43,716 telemetry samples, where each telemetry sample has a feature vector of 8 different statistical quantities, as discussed in Section 3. With these telemetries, we construct seven 3rd-order telemetry tensors, and the dimensions and the size of each tensor are provided as follows: $time \times feature\ vector \times$

*telemetry*, and $43,716 \times 8 \times number\ of\ telemetries$. We summarize the detailed size for each subsystem in Table 2.

**Convergence Criteria:** Next, we decompose each 3rd-order telemetry tensor into component matrices, $A$, $B$, and $C$, using CANDECOMP/PARAFAC (CP) decomposition with the alternating least squares (ALS) updating method. Updating will be stopped when the iteration reaches 100, or when the convergence tolerance is less than $10^{-8}$ in this experiment.

**Optimal Rank:** The reconstruction errors are calculated from increasing the rank $r$ from 2. Then, we choose the smallest $r$, which minimizes $\|X - \hat{X}\|_{\mathbf{F}}$, until when the approximate tensor can reconstruct more than 90% of the original telemetry tensor. The result of the optimal $r$ is presented in the third column in Table 2 for each subsystem. As shown in Table 2, $r$ produces different ranges of values from 11 to 29 because of the different telemetry values, characteristics, and structures in each subsystem.

Table 2: The size of 3rd-order telemetry tensor, the optimal rank $r$ where every $r$ guarantees more than 90% of the reconstruction rate of an estimated tensor, and the optimal number of clustering $k$ by silhouette analysis for each subsystem.

| Subsystem | Tensor size $(time \times f.vec. \times tel.)$ | Optimal rank $r$ | Optimal $k$ |
|---|---|---|---|
| *subsystem1* | $43,716 \times 8 \times 7$ | 20 | 3 |
| *subsystem2* | $43,716 \times 8 \times 9$ | 28 | 7 |
| *subsystem3* | $43,716 \times 8 \times 14$ | 20 | 5 |
| *subsystem4* | $43,716 \times 8 \times 26$ | 29 | 2 |
| *subsystem5* | $43,716 \times 8 \times 15$ | 20 | 5 |
| *subsystem6* | $43,716 \times 8 \times 10$ | 18 | 2 |
| *subsystem7* | $43,716 \times 8 \times 7$ | 11 | 9 |

***k*-means Clustering:** Among the three-component matrices from decomposition using the optimal $r$, we apply $k$-means clustering to the component matrix $A$ consisting of time-to-factor information. Since our goal is to detect anomaly points over different time values, we chose the component matrix $A$, which has the time information. Additionally, in order to determine an optimal $k$, we apply silhouette analysis for each subsystem. All results are presented in the last column in Table 2.

**Fine-Tuning for Dynamic Thresholding:** We use the dynamic thresholding method, which can dynamically adjust the threshold value based on environmental change. However, there are clear trade-offs between different dynamic thresholding parameters, window size $w$ and coefficient parameter $m$. To empirically evaluate the trade-offs and fine-tune the best parameters, we conducted various experiments by changing the window size from 9 to 576 to determine the optimal value of $w$ and $m$ for each subsystem. Since we sampled the dataset corresponding to 10 minutes into one data point, window size 9 translates to 90 minutes time period, which corresponds to the single activity cycle of the satellite operation. We can observe that the best performance is achieved when $w$ is either 108 or 198 for all subsystems. We also empirically found that $m = 6$ typically resulted in the best performance for any window size. An example of fine-tuning for dynamic thresholding is described in Appendix C.

**Comparisons with Other Methods:** We compared our approach with four other well-known anomaly detection baselines developed for satellite systems. First, we compare ours with One-Class SVM (OCSVM) after feature extraction following the method from CNES's NOS-TRADAMUS (Fuertes et al., 2016) approach. Second, Isolation Forest (IF) is used instead of OCSVM. The Next approach is based on Mixtures of Probabilistic Principal Component Analysis and Categorical Distributions (MPPCACD) model used for JAXA (Yairi et al., 2017)'s SDS-4. Lastly, a single-channel LSTM (SC-LSTM) model developed by NASA-JPL (Hundman et al., 2018) is also compared.

## 6 RESULTS

We present the performance of each anomaly detection method in Table 4. The domain experts label anomalies at KARI, and the number of them is shown in the second column in Table 3. If a detected point is an anomaly, it is counted as a true positive (TP). Otherwise, we count it as a false positive (FP). The performance in TP and FP for each detection method is provided in Table 3.

*Note*: Even with 10 months of data, we do not have many anomalies. If there are many anomalies, the satellite will not function properly. Specifically, as requested by satellite operators, our objective is to detect the anomalous events accurately, while reducing false positives in a highly unbalanced dataset.

Table 3: Anomaly labels, and the performance comparison of TP and FP with different anomaly detection methods.

| Subsystem | Anomaly | MPPCACD (JAXA) | | SCLSTM (NASA) | | OCSVM (CNES) | | IF | | ITAD (Ours) | |
|---|---|---|---|---|---|---|---|---|---|---|---|
| | | TP | FP | TP | FP | TP | FP | TP | FP | TP | **FP** |
| *Subsystem1* | 0 | 1 | 0 | 1 | 0 | 0 | 0 | 0 | 0 | 1 | 0 |
| *Subsystem2* | 0 | 1 | 1 | 1 | 0 | 2 | 0 | 2 | 0 | 1 | 1 |
| *Subsystem3* | 1 | - | 4 | - | 10 | - | 1 | - | 1 | - | 0 |
| *Subsystem4* | 1 | - | 1 | - | 10 | - | 12 | - | 12 | - | 0 |
| *Subsystem5* | 2 | - | 3 | - | 5 | - | 5 | - | 5 | - | 0 |
| *Subsystem6* | 0 | - | 0 | - | 0 | - | 0 | - | 0 | - | 0 |
| *Subsystem7* | 0 | - | 0 | - | 0 | - | 0 | - | 0 | - | 0 |
| Total | 3 | 2 | 9 | 2 | 25 | 2 | 18 | 2 | 18 | 2 | **1** |

All methods show the same detection performance in TP while detecting anomalies from different subsystems. However, our ITAD and MPPCACD detect one anomaly for *subsystem1* and another one for *subsystem2*. OCSVM detects all two anomalies from *subsystem2*, while it misses an anomaly for *subsystem1*. On the other hand, the ITAD framework outperforms in FP compared to others. While SCLSTM, OCSVM, and IF produces a total of 25, 18, and 18 false positives, respectively, our ITAD approach reduces the false positives to 1. MPPCACD shows the second-best performance among other approaches in FP, but it could not reduce the FPs enough compared to the ITAD approach.

To compare the overall performance, we present the precision, recall, and F1 score in Table 4. The ITAD framework achieves the highest precision (66.67%) than other methods, because of its high performance in FP. On the other hand, the recall performance was the same across all approaches. Overall, ITAD outperforms all other methods in F1 score by more than two-fold (66.67% vs. 28.57%).

Table 4: Performance comparison of ITAD with other methods.

| Method | Precision | Recall | F1 Score |
|---|---|---|---|
| MPPCACD (JAXA) | 7.41% | 66.67% | 13.33% |
| SCLSTM (NASA) | 18.18% | 66.67% | 28.57% |
| OCSVM (CNES) | 10.00% | 66.67% | 17.39% |
| IF | 10.00% | 66.67% | 17.39% |
| **ITAD (Ours)** | **66.67%** | 66.67% | **66.67%** |

**Analysis:** Most of the anomaly detection methods except our approach generate high FPs because they cannot account for multiple telemetries simultaneously. When a temporary glitch is detected from only one telemetry in a subsystem, it is highly likely that it is a trivial outlier, not an actual anomaly. (Note that telemetries collected from adjacent sensors in the same environment are regarded as one case since they are highly correlated with each other). In the case of *subsystem4*, as shown in Fig. 2.(a), the 1st telemetry, *TDCSUAT*, has a temporal glitch on August 17th, whereas all other different types of telemetries such as *MMQTXON*, *TCCU2AT*, and *TXPOND2T* do not have any glitch at the same timestamp. This glitch is confirmed as a trivial outlier, not an actual anomaly.

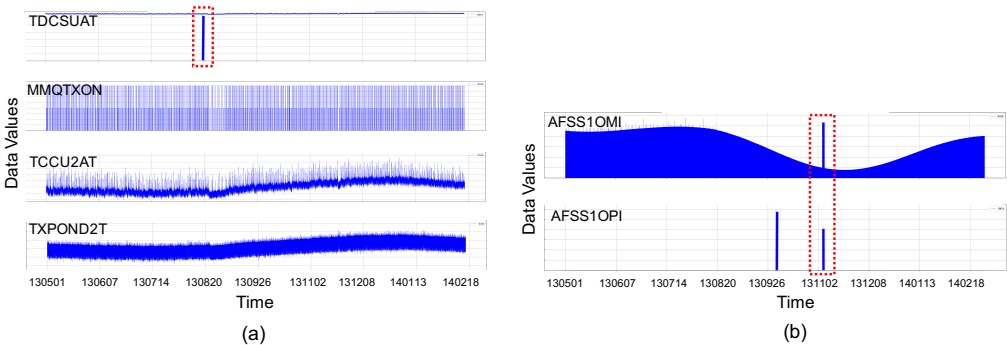

Figure 2: (a) Plots of different types of telemetries (*TDCSUAT*, *MMQTXON*, *TCCU2A* and *TXPOND2T*) in *subsystem4*. A temporal glitch is not detected anywhere except *TDCSUAT*. According to the verification by an expert, this temporal glitch is a trivial outlier, not an actual anomaly. (b) Plots of different types of telemetries (*AFFS1OMI* and *ASFF1OPI*) in *subsystem2*. Temporal glitches are discovered from different telemetries at the same timestamp. According to the verification by an expert, they are actual anomalies.

Our Integrative Tensor-based Anomaly Detection (ITAD) approach does not report this glitch as an anomaly, whereas other detection methods such as MPPCACD, SCLSTM, OCSVM, and IF record it as an anomaly. In the case of *subsystem3*, *subsystem4*, and *subsystem5*, there are some temporal glitches, but no actual anomaly. The ITAD framework is the only method that does not report this type of trivial outliers, as shown in FPs of *subsystem4* in Table 3.

When there are temporary glitches in multiple telemetries at the same timestamp as shown in *subsystem2* in Fig. 2.(b), ITAD reports anomalies. For example, there are temporary glitches in two different types of telemetries (*AFFS1OMI* and *ASFF1OPI*) at the same timestamp. This glitch is an actual anomaly, and the ITAD approach accurately reports it as an anomaly. Since the ITAD method can take and process multiple telemetries simultaneously, it significantly reduces false positives caused by the other methods based on a single-variate anomaly analysis. These results demonstrate the effectiveness of integrative analysis for multiple telemetries in subsystems using a tensor-based method for satellite monitoring.

## 7   Discussion and Limitations

Determining an appropriate rank-size $r$ is an *NP*-complete problem (Håstad, 1990), and there is no general algorithm to find it. To choose $r$, we exploit the reconstruction error, which is proposed in the original CP research (Carroll and Chang, 1970; Harshman, 1970). However, there is a possibility to suffer from overfactoring and ultimately failing to obtain an optimal solution from this method. To address this possibility, we plan to apply the Core Consistency Diagnostic (CORCONDIA) proposed by Bro and Kiers (2003) for determining the optimal rank $r$ for our future work. We believe that the CORCONDIA method, which assesses the core consistency and measures the similarity between the core array and theoretical super-diagonal array, can yield more accurate results.

Even though we use 10 months of real telemetry dataset, we do not have many anomalies, which is a realistic scenario. Otherwise, i.e. if there are many anomalous events, most mission-critical systems would fail very quickly. In the presence of a small number of anomalies, the main focus of our work is to reduce false positives to assist satellite operators to determine the true anomalies, as requested by KARI operators. However, we agree that because of a small number of anomalies, current precision, and recall metrics would be very sensitive to anomaly events. Missing one anomaly would result in a 33% drop in performance. To partially address this issue, we are currently in the process of collecting more datasets with anomalies within a longer and plan to evaluate our tensor-based system with datasets with more anomalies. Also, we believe we need to develop a better performance metric, which can capture the performance with a small number of anomalies.

Lastly, we are in the process of deploying our tensor-based anomaly detection method to THE KOMPSAT-2 satellite in the spring of 2020. We plan to incorporate not only 88 telemetries we experimented in this research, but also other types of telemetries and subsystems to evaluate our integrative anomaly detection method.

## 8   Conclusion

In this work, we proposed an Integrative Tensor-based Anomaly Detection framework (ITAD) to detect anomalies using the KOMPSAT-2 satellite telemetry dataset, where our approach can analyze multiple telemetries simultaneously to detect anomalies. Our ITAD achieves higher performance in precision and F1 score compared to other approaches. We also demonstrate that the ITAD reduces the false positives significantly. This reduction in FPs is because it can distinguish actual anomalies from trivial outliers by incorporating information from other telemetries at the same time. In the future, we plan to improve our algorithm by applying the CORCONDIA method to avoid overfactoring and find an optimal rank $r$ and incorporate and evaluate datasets with more anomalies.

We believe our work laid the first grounds using an integrated tensor-based detection mechanism for space anomaly detection. Moreover, the result demonstrates that our proposed method can be applicable in a variety of multivariate time-series anomaly detection scenarios, which require low false positives as well as high accuracy.

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

# A    TENSOR DECOMPOSITION

## A.1    A TENSOR

A tensor is a multi-dimensional array (Papalexakis et al., 2017), where geometric vectors and scalars can be considered as the simplest tensors. A 1st-order tensor is a vector, a 2nd-order tensor is a matrix, and a 3rd-order tensor can be represented as a cube, which has three vector spaces. In general, $N$th-order tensor $\boldsymbol{X} \in \mathbb{R}^{I_1 \times I_2 \times \cdots \times I_N}$ is represented by the outer product $\circ$ of $N$ vector spaces as follows:

$$\boldsymbol{X} = a^1 \circ a^2 \circ \cdots \circ a^N, \tag{3}$$

where $I_N$ is the $N$th dimension and $a^N$ is the vector in $N$th dim. A rank in a tensor indicates the number of components in the decomposed matrices and every tensor can be expressed as a sum of a rank-1 tensor (Kruskal, 1977; 1989). Due to its ability to express multi-modality, it is effective to handle such dataset with multi-modal aspects. Expressing a tensor as a sum of a rank-1 tensor was first proposed by Hitchcock (1927; 1928).

## A.2    TENSOR DECOMPOSITION

In 1970, Carroll and Chang (1970) proposed a canonical decomposition (CANDECOMP) and Harshman (1970) suggested parallel factor decomposition (PARAFAC), which is an extended version of 2-way factorization for higher-order data. Since CANDECOMP and PARAFAC have a similar concept, the CANDECOMP/PARAFAC (CP) decomposition formulated by Kiers (2000) has been widely used. It decomposes $N$th order data into a linear sum of rank-1 tensor as described in Fig. 3.(a) and a 3rd-order tensor can be decomposed into three component matrices $A$, $B$, and $C$. Appellof and Davidson (1981) pioneered the use of CP model to extract information from a chemical system. Andersen and Bro (2003) contributed to developing practical description and application of tensors. And Acar et al. (2005; 2006) was the first to apply a tensor decomposition to data mining. They analyzed online chatroom data to understand how social groups evolved in cyberspace. They constructed a 3rd-order tensor with $user \times keyword \times time$ spaces.

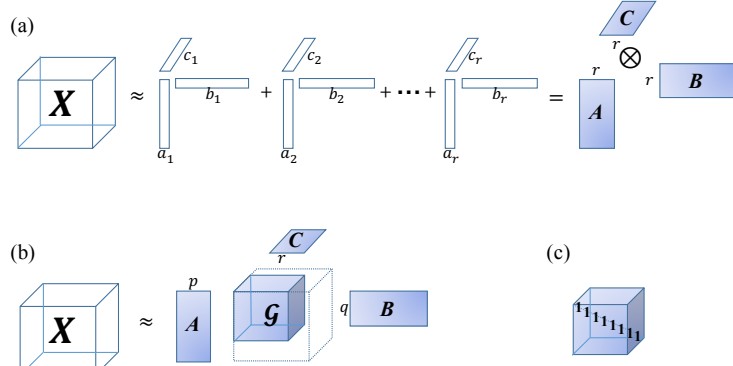

Figure 3: Examples of different tensor decomposition methods from the given 3rd-order tensor: (a) CANDECOMP/PARAFAC (CP) decomposition, where $\boldsymbol{X}$ is an approximated sum of $r$ rank-1 tensors. (b) Tucker decomposition, where $\boldsymbol{X}$ has three component matrices with the a rank-(p, q, r) and a core tensor g. (c) A super-diagonal core tensor with ones as diagonal values. If the core tensor of Tucker decomposition is a super-diagonal having ones as diagonal values and all ranks are identical, it can be thought as same as CANDECOMP/PARAFAC decomposition.

The Tucker decomposition is another commonly-used tensor decomposition, which was first introduced by Tucker (1963) and refined later (Tucker, 1964; 1966). As shown in Fig. 3.(b), the Tucker decomposition has a core tensor, which can be viewed as a compression of the original tensor $\boldsymbol{X}$. In the case of the 3rd-order tensor, it decomposes a tensor into a core tensor and three matrices with different ranks ($p \neq g \neq r$) as shown in Fig. 3.(b). In fact, CP decomposition can be thought as a special case of the Tucker decomposition, where all ranks are identical ($p = g = r$) and the core tensor is super-diagonal having ones as diagonal values as shown in Fig. 3.(c). De Lathauwer and Vandewalle (2004) applied the Tucker decomposition for dimensionality reduction in higher-order

signal processing. Pioneers of the use of the Tucker decomposition in computer vision are Vasilescu and Terzopoulos (2002a). They extended conventional Singular Values Decomposition (SVD) to $N$-mode SVD for a tensor. They also showed that it has significantly better performance compared to the standard Principal Component Analysis (PCA) for image recognition task (Vasilescu and Terzopoulos, 2002b).

The benefit of using tensor decomposition is that it is one of the most effective unsupervised methods for extracting characteristics of $N$th-dimensional data. Traditionally, it has been required to rearrange the dimension into a 2nd-order matrix to factorize high dimensional data. However, tensor decomposition can offer more accurate results by keeping the $N$th-order structure of data as shown in other research (Acar and Yener, 2009; Kolda and Bader, 2009; Cichocki et al., 2015).

## B    SELECTING OPTIMAL PARAMETERS

### B.1    OPTIMAL $k$

We use silhouette analysis to determine an optimal $k$ for $k$-means clustering. We varied $k$ from 2 to 10, and chose the value when the silhouette coefficient is the largest positive value as shown in the example of *Subsystem2* in Fig. 4.

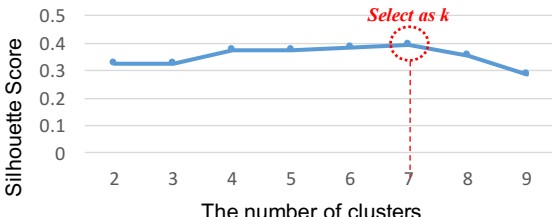

Figure 4: Selecting the optimal $k$ for *Subsystem2* using the silhouette scores by changing the number of clusters.

### B.2    OPTIMAL $r$

In order to select the optimal $r$, we find the reconstruction error by increasing the rank $r$ from 2 and choosing the smallest $r$ by minimizing $\|X - \hat{X}\|_{\mathbf{F}}$, until when the approximated tensor can reconstruct more than 90% of the original telemetry tensor as shown in the example of *Subsystem7* in Fig. 5.

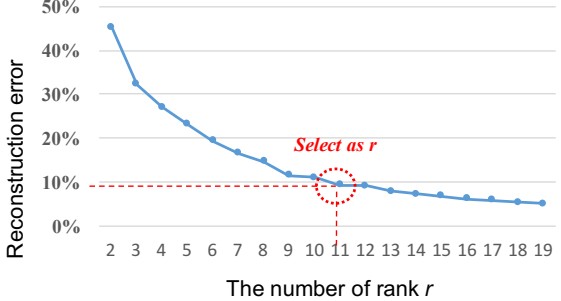

Figure 5: Selecting the optimal $r$ for *Subsystem7* at the point when the reconstruction error is less than 10%.

## C    FIXED AND DYNAMIC THRESHOLDING

### C.1    PROBLEM WITH FIXED THRESHOLDING

A fixed threshold approach cannot adapt to changing conditions, and can produce too many false positives. An example of the problems with a fixed threshold and high false positives is illustrated in Fig. 6.

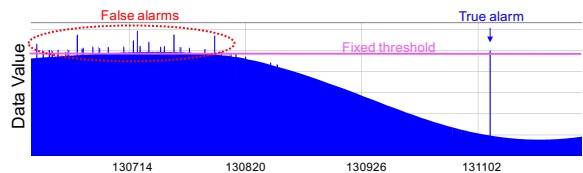

Figure 6: Example of high false alarms caused by fixed threshold for telemetry *AFSS1OMI* of *subsystem2*.

## C.2 FINE-TUNING FOR DYNAMIC THRESHOLDING

The X-axis indicates the time, and the Y-axis is the anomaly score value. The blue line indicates the anomaly score and the red line represents the dynamic threshold computed from the formula, $\mu + m \cdot \sigma$ during $w$. As we can observe from Fig. 7, different number of false positives and true positives can be detected based on different $w$ and $m$. As shown in Fig. 7.(a) and (b), increasing the window size $w$ from 9 to 198 tends to make the threshold line flatter. In addition, Fig. 7. (a) and (c) show that increasing the coefficient parameter $m$ from 4 to 6 influences the overall distance between the threshold line and the anomaly score line. We can observe the best performance is achieved when $w$ is either 108 or 198 for all subsystems. We also empirically found that $m = 6$ typically resulted in the best performance for any window size.

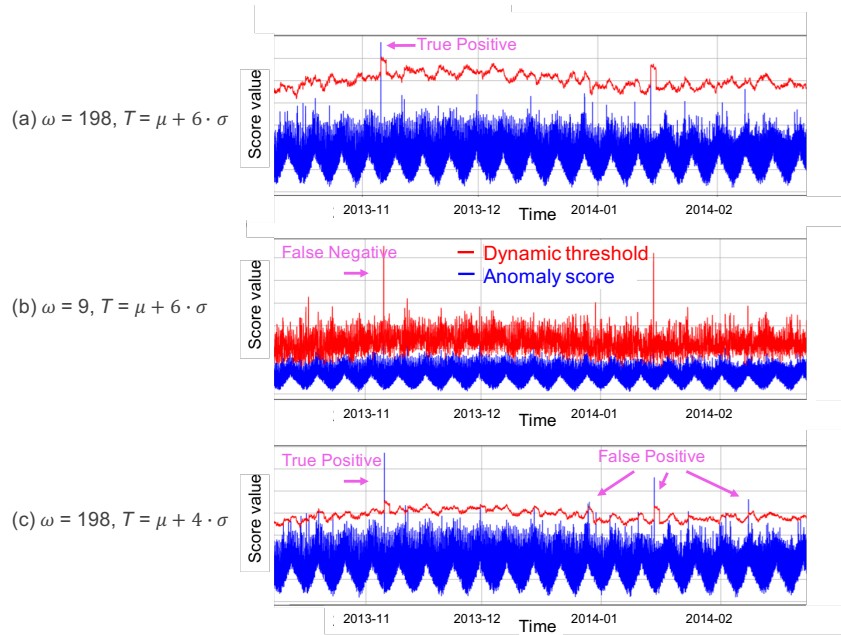

Figure 7: Anomaly score (blue line) and dynamic threshold (red line) by the size of the time window ($w$) and coefficient parameter ($m$): (a) when $w = 198$, $T = \mu + 6 \cdot \sigma$, it detects actual anomaly only (1 true positive), (b) when $w = 9$, $T = \mu + 6 \cdot \sigma$, it cannot detect any (1 false negative), and (c) when $w = 198$, $T = \mu + 4 \cdot \sigma$, it detects many anomalies including the actual anomaly (1 true positive and 3 false negatives). (a) is optimal, (b) is too loose, and (c) is too sensitive.

