# OpenReview forum: "Integrative Tensor-based Anomaly Detection System For Satellites"
_ICLR.cc/2020/Conference — Reject_

### Official Review · AnonReviewer1 · 2019-10-15
**Official Blind Review #1**

**Rating:** 1

**Review:**

In this paper, a tensor-based anomaly detection system for satellites is proposed. The proposed method mainly consists of three parts: creating a tensor from time-series data, applying tensor decomposition, and clustering. The performance is evaluated with a real satellite data set.

Though the results look interesting, I vote for rejection because the paper is not well fitted to the ICLR community.

This paper specifically focuses on the anomaly detection of satellites. I don't say the application study is less worthy but the scope of this paper is too narrow for the ML community. The methodology of this work is basically combining the existing technology and is not novel enough. The performance is validated only by a single data set. So I feel this paper is more like a case study of a specific application and is more appropriate submitting to data mining conferences such as KDD.

**Experience Assessment:**

I have published one or two papers in this area.

**Review Assessment: Checking Correctness Of Derivations And Theory:**

N/A

**Review Assessment: Checking Correctness Of Experiments:**

I assessed the sensibility of the experiments.

**Review Assessment: Thoroughness In Paper Reading:**

N/A

---

### Official Review · AnonReviewer2 · 2019-10-24
**Official Blind Review #2**

**Rating:** 3

**Review:**

The paper presents and evaluates an anomaly detection algorithm that is used for identifying anomalies in satellite data. The considered satellite data originates from multiple sensors that are divided into sensor groups. The anomaly detection algorithm roughly works as a 4 stage process:
Raw data is converted into a set of features per sensor that aggregate the last 10 min of sensor data, resulting in a tensor X with dimensions (time, features, sensors)
A tensor decomposition method is used to decompose X into matrices A,B,C.
Matrix A with dimensions (time, latent factors) is used to cluster data using k-means.
Time points are considered anomalous when the latent feature vector is too far away all cluster centres.
This algorithm is experimentally compared to baseline methods that only use one sensor at a time.

Overall, I think this paper is not fully ready for publication based on a) inconsistencies in notation that make it rather hard to read at first and b) open questions regarding the baselines, and c) missing experimental evaluations that may give insights into why the algorithm works better then the baselines. Moreover, besides those shortcomings, d) I am not convinced that ICLR is the right venue for this paper. Although the data set could provide an interesting application, the paper provides no theoretical contributions and the algorithm only pipelines known algorithms in a problem-specific way. In my view, a more application-focussed conference would be a better fit.


Detailed comments.

d) in my view is the most important issue. As mentioned above, the paper contains no theoretic contributions to the field of anomaly detection. Instead, it solves data-specific challenges for anomaly detection by pipelining a variety of methods in order to predict anomalies. While this approach is fine for solving an application, I am not convinced that it warrants a publication at ICLR, regarding in particular that the experimental evaluation does not provide any insights into why the algorithm works better than the baselines. That is,
c) I am missing experiments that show the contributions of the individual choices that were made for each stage of the detection process. E.g. how much influence does the adaptive thresholding have on the performance? (Appendix C doesn’t quantify that properly.) The same would be interesting for the other stages.

b) You state that the baseline methods are single-telemetry-based. However, you do not explain how those methods detect anomalies for multi-telemetry data as presented in Tables 3,4. Do those methods detect an anomaly whenever any of the feature/telemetry-specific methods detect one?
Also, Table 3 is not correct and doesn’t match the analysis in the text. Labels do not seem to be correct. Does Subsystem1 have 1 anomaly and Subsystem2  2?

a)
-P4: Indices for the 3 dimensions of input data X are sometimes denote I_1,I_2,I_3 and sometimes I,J,K.
-P4: the rank or number of factors is sometimes denied by lowercase r and sometimes by uppercase R, e.g. in the definition of the reconstruction error. Here, the running variable is denoted by lowercase r.
-P5, ‚Calculating anomaly score‘: Shouldn’t time samples have r dimensions instead of n?
-P5, last paragraph of Section 4: \mu should be defined by the normaliser w instead of n.

**Experience Assessment:**

I have read many papers in this area.

**Review Assessment: Checking Correctness Of Derivations And Theory:**

I assessed the sensibility of the derivations and theory.

**Review Assessment: Checking Correctness Of Experiments:**

I assessed the sensibility of the experiments.

**Review Assessment: Thoroughness In Paper Reading:**

I read the paper at least twice and used my best judgement in assessing the paper.

---

### Decision · Program_Chairs · 2019-12-19

**Decision:**

Reject

**Comment:**

The paper applies tensor analysis techniques to anomaly detection from satellite data. The proposed solution is simple and seems to achieve good results. However, there is limited novelty in methodology and no sufficient experiments have been conducted to explain the performance gain. The paper is not ready for publication in ICLR but could be suitable for an application oriented venue.